# Extensive nrDNA Polymorphism in *Morus* L. and Its Application

**DOI:** 10.3390/plants14162570

**Published:** 2025-08-18

**Authors:** Xiaoxiang Xu, Le Zhang, Changwei Bi, Meiling Qin, Shouchang Wang, Dong Li, Ningjia He, Qiwei Zeng

**Affiliations:** 1State Key Laboratory of Resource Insects, Institute of Sericulture and Systems Biology, Southwest University, 216 Tiansheng Road, Beibei District, Chongqing 400716, Chinalidong203@swu.edu.cn (D.L.); 2State Key Laboratory of Tree Genetics and Breeding, Co-Innovation Center for Sustainable Forestry in Southern China, Key Laboratory of Tree Genetics and Silvicultural Sciences of Jiangsu Province, Nanjing Forestry University, Nanjing 210037, China; bichwei@njfu.edu.cn

**Keywords:** *Morus*, nrDNA, polymorphism, ITS, taxonomy, CAPS

## Abstract

The internal transcribed spacer (ITS) is one of the most extensively utilized in the taxonomy of the genus *Morus* due to its generally concerted evolution. Although non-concerted evolution of nuclear ribosomal DNA (nrDNA) has been reported in some species, genome-wide nrDNA characteristics in the genus *Morus* remain poorly understood. In this study, 158 single-nucleotide polymorphisms (SNPs) and 15 insertions and deletions (InDels) were identified within the nrDNA regions of 542 mulberry accessions representing sixteen *Morus* species. These wide occurrences of heterogeneous SNPs and InDels revealed the intra-individual polymorphism within the nrDNA region of *Morus*, indicating the incomplete concerted evolution of nrDNA. Notably, 66 out of 158 SNPs and 13 out of 15 InDels were localized within the ITS regions (ITS1-5.8S-ITS2), indicating a high degree of polymorphism in the ITS, which was further validated through classical cloning and Sanger sequencing methodologies. The 13/16 bp InDel located in the ITS1 region was utilized to develop a rapid and reliable cleaved amplified polymorphic sequence (CAPS) marker-based method for distinguishing *M*. *alba* and *M. notabilis* from other *Morus* species, eliminating the need for a clone-based sequencing step or comparative phenotypic analysis. Phylogenetic analysis based on nrDNA SNPs from 542 mulberry accessions revealed six distinct clades, corresponding to the six *Morus* species. These findings offer novel new insights into the taxonomy, conservation, and breeding improvement of *Morus* species.

## 1. Introduction

The nuclear ribosomal DNA (nrDNA) array in higher plants comprises hundreds of tandem repeat units, each encoding three conserved ribosomal RNA genes (18S, 5.8S, and 26S), which are separated by two internal transcribed spacers (ITS1 and ITS2) and flanked by an external transcribed spacer (ETS) and an intergenic spacer region (IGS) [1,2,3,4]. The internal transcribed spacers, together with the 5.8S gene, form the ITS (ITS1-5.8S-ITS2) region [5]. The ITS region has been widely employed in phylogenetic studies for species identification, species delimitation, and DNA barcoding due to its concerted evolution within and between constituent subunits, its rapid evolution, its short length, and the availability of universal primers [6,7,8]. Concerted evolution results in the homogenization of nrDNA tandem repeats, leading to lower ITS intraspecific variability in the ITS relative to interspecific variation [3]. Owing to the concerted evolution of the ITS [9], intra-individual polymorphism has traditionally been regarded as an exception [6,10]. However, advances in sequencing technologies have uncovered intra-individual ITS polymorphisms across a broad spectrum of species [6,11,12,13,14].

Mulberry (*Morus* L.), exhibiting ploidy levels ranging from diploid (2n = 2x) to docosaploid (2n = 22x) [15,16], belongs to the order Rosales, family Moraceae, and genus *Morus* [17]. Its heterozygosity and open-pollination behavior have resulted in numerous natural and artificial hybrids and polyploids, significantly complicating its genetic background and taxonomy. Since Linnaeus first described seven *Morus* species [18], the taxonomy of the genus *Morus* has undergone extensive revisions (ranging from 5 to 35 accepted species) based on morphological and molecular methods [19,20,21,22,23,24,25,26,27,28]. *Morus mongolica*, *M. cathayana*, *M. australis*, *M. wittiorum*, *M. indica*, *M*. *atropurpurea*, *M. bombycis*, and *M. macroura* were proposed as *M. alba* species using the ITS marker and chloroplast genomes [20,25]. Population genetic analyses have demonstrated that traditional morphology-based taxonomy does not accurately reflect the true phylogenetic relationships within the genus [29]. However, recent studies have treated *M. cathayana*, *M. indica*, *M. macroura*, *M. mongolica*, *M. wittiorum*, and *M. alba* as distinct species [26,27,28]. Additionally, *M*. *alba* var. *atropurpurea* has been proposed as a separate species, *M. atropurpurea* [30]. These discrepancies highlight the need for further taxonomic revision through comprehensive analysis by extensive sampling. Fortunately, recent advances in mulberry genomics, including multiple resequenced genomes and reference genomes [29,30,31,32,33], provide a robust foundation for genome-level taxonomic revisions within the *Morus* species.

Over the past 5000 years, *M*. *alba* has been domesticated for the production of high-quality leaves and fruits, resulting in numerous cultivated varieties. More than 5220 mulberry germplasm resources are maintained, primarily in China (over 2600), Japan (1500), and India (1291) [34,35]. *M. alba* spread from Asia to Europe, America, and Africa via the Silk Road [36], and its leaves have been used for sericulture in over 70 countries [34]. Following its introduction to the United States during colonial times, *M. alba* expanded across North America [37]. This has rendered *M*. *alba* an invasive species, as it hybridizes with and displaces the indigenous red mulberry (*M. rubra*), leading to the listing of *M*. *rubra* as an endangered species in Canada [38,39,40,41]. These factors complicate the identification of *M. alba*.

As a popular molecular marker, the ITS has been employed in comparative analyses for the taxonomy of *Morus* [20,42,43,44]. Despite the deposition of three hundred and seventy-four *Morus* ITS sequences in GenBank, only three partial 18S sequences and two partial 26S sequences have been isolated from *Morus* (up to July 2024). Consequently, the characteristics of nrDNA regions, particularly at genome level and across diverse *Morus* accessions, remain poorly understood, necessitating further investigation. The CAPS assay is a robust and straightforward method for detecting genetic polymorphism, involving the digestion of amplified DNA fragments with restriction enzymes [45,46]. This method has been utilized for cultivar identification in various species, including *Arabidopsis* [45], *Fragaria* × *ananassa* [47], *Brassica napus* [48,49], *Linum usitatissimum* [50], and *Capsicum annuum* [51]. Furthermore, an ITS-based CAPS has been developed for differentiating wild native lilies [46].

Therefore, the present study aims to (1) systematically identify and comprehensively analyze the characteristics of the nrDNA region in the *Morus* genus and (2) reconstruct the phylogenetic tree of the *Morus* genus using the SNPs of nrDNA regions. We also provide a taxonomic revision of *Morus* based on pattern profiles of SNPs and InDels of the nrDNA regions. This study also develops a simple and efficient CAPS marker-based method for distinguishing *M. alba* and *M. notabilis* germplasm resources.

## 2. Results

### 2.1. SNPs and InDels in nrDNA of Morus

Despite numerous *Morus* ITS sequences being available in GenBank, comprehensive genomic data on *Morus* nrDNA remain limited, necessitating further investigation. In this study, a total of 542 mulberry accessions, representing sixteen *Morus* species (Table 1), were analyzed using GATK software (version 4.6.1.0) to identify nrDNA variants. Among these, 30 accessions belonged to *M. notabilis*, *M. celtidifolia*, *M. nigra*, *M. rubra*, and *M. serrata*, while the remaining accessions represented *M. alba*, *M. bombycis*, *M. australis*, *M. cathayana*, *M. macroura*, *M. mongolica*, *M. wittiorum*, *M. latifolia*, *M. mizuho*, *M. liboensis*, and *M. rotundiloba* (Table 1). The average depth of coverage for the nrDNA region was less than 400-fold for six accessions, while the remaining five hundred and thirty-six accessions exhibited coverage ranging from 2625- to 108,186-fold (Appendix A). A total of 158 high-quality SNPs and 15 InDels were identified in the nrDNA regions of *Morus*, displaying unique patterns corresponding to six *Morus* species including *M. nigra*, *M. serrata*, *M*. *rubra*, *M. celtidifolia*, *M. alba*, and *M. notabilis* (Figure 1, Appendix A), revealing interspecific diversity and intraspecific conservation.

Notably, 66 SNPs and 13 InDels were detected within the ITS regions, exhibiting a high level of ITS polymorphism in *Morus*. Specifically, 77, 47, 44, 43, 42, and 23 SNPs were detected in *M. notabilis*, *M*. *rubra*, *M. serrata*, *M. nigra*, *M. celtidifolia*, and *M. alba*, respectively (Figure 1A, Table 2). In the 18S region, six SNPs were detected in 31 *M. alba* accessions, while six, eight, eight, six, and eleven SNPs were identified in the 28 accessions of *M. nigra*, *M. rubra*, *M. serrata*, *M. celtidifolia*, and *M. notabilis*, respectively (Figure 1A, Table 2 and Appendix A). In the 5.8S region, three SNPs were documented in three *M*. *rubra* accessions, one SNP in all *M*. *notabilis* accessions, one SNP in seven *M*. *alba* accessions, and one SNP in one *M*. *serrata* accession. In the 26S region, 10 SNPs were found in 29 *M. alba* accessions, and 20, 23, 17, 17, and 32 SNPs were identified in the 28 accessions of *M. nigra*, *M. rubra*, *M. serrata*, *M. celtidifolia*, and *M. notabilis*, respectively. Among 514 *M. alba* accessions, 23 *M. alba* SNPs were detected in 57 accessions (including *M. alba*, *M. australis*, *M. wittiorum*, *M. bombycis*, *M. latifolia*, *M. mongolica*, *M. macroura*, and *M. cathayana*), while the remaining 457 accessions exhibited no SNPs, indicating identical nrDNA to the *M. alba* reference sequence (Figure 1A, Table 2, and Appendix A). Although 72 out of 77 SNPs identified in 16 *M. notabilis* accessions were homogeneous, 4 SNPs (7, 28, 51, and 75) from 2 *M. yunnanensis* accessions and 1 SNP (34) from 13 *M. notabilis* accessions were heterogeneous (Figure 1A, Table 3). Additionally, 44 *M. serrata* SNPs, 43 *M. nigra* SNPs, 42 *M. celtidifolia* SNPs, and 47 *M. rubra* SNPs comprised 3, 6, 6, and 17 heterogeneous SNPs, respectively, with the remained being homogeneous (Table 3).

Eight distinct InDels were identified in the sixteen *M. notabilis* accessions and three *M. nigra* accessions, whereas six polymorphic InDels were detected in five *M. celtidifolia* accessions, three *M. rubra* accessions, and one *M*. *serrata* accession, respectively (Figure 1B, Table 2). Three InDels were found in 28 out of 514 *M. alba* accessions (*M*. *alba*, *M*. *australis*, *M*. *bombycis*, and *M*. *wittiorum*). Among the 28 accessions, the *M. macroura* (SRR26434445) and *M. wittiorum* accessions (CNR0342481) contained two homogeneous InDels, while 5 and 21 accessions exhibited two and one heterogeneous InDels, respectively. Furthermore, two and one heterogeneous InDels were identified in three *M. nigra* accessions and two *M. yunnanensis* accessions, respectively, with all other InDels in *M. nigra*, *M. celtidifolia*, *M. serrata*, *M. rubra*, *M. notabilis*, and *M. yunnanensis* being homogeneous (Table 3 and Appendix A). Thirteen InDels were located in the ITS1 and ITS2 regions, while one InDel was detected in the 26S region of three *M*. *nigra* accessions and two *M*. *yunnanensis* accessions. No InDels were identified in the 5.8S or 18S regions. The largest InDel, consisting of either a 13- or 16-base-pair insertion (“CGTGCGCAATGCG” or “CGACGTACACAATGCG”) (Figure 1B, Appendix A), was mapped to the nucleotide position 1862 of nrDNA regions in 35 out of 542 mulberry accessions. Homogeneous 13 bp InDels were detected in five *M. celtidifolia* accessions, three *M. nigra* accessions, three *M. rubra* accessions, one *M*. *wittiorum* accession, one *M. macroura* accession, and one *M. serrata* accession, whereas homogeneous 16 bp InDels were exclusively present in sixteen *M*. *notabilis* accessions. Notably, a heterogeneous 13 bp InDel was observed in three *M*. *australis* accessions and one hybrid offspring derived from *M*. *australis* and *M*. *alba*. Although 13/16 bp InDels were identified in only 35 accessions, raw sequencing data revealed 539 accessions containing both 13/16 bp insertion and reference nucleotides (Appendix A). The widespread occurrences of heterogeneous SNPs and InDels in *Morus* accessions revealed the intra-individual polymorphism of *Morus* nrDNA, suggesting incomplete concerted evolution of nrDNA.

### 2.2. Inter- and Intraspecific Polymorphism of the nrDNA Region in Morus and Phylogenetic Analysis

To comprehensively analyze nrDNA polymorphism among 542 mulberry accessions, SNP data were converted into 542 FASTA sequences, from which 31 unique sequences (Figure 2A) were obtained by removing repetitive sequences using the seqkit (version 2.5.0) software (Appendix A). These repetitive sequences were categorized into nine different groups, comprising 457, 21, 13, 13, 6, 4, 2, 2, and 2 sequences (Figure 2A). The analysis revealed extensive interspecific SNP variation in *Morus*, with limited intraspecific polymorphism. The largest group consisted of 457 sequences identical to the reference sequence, including 243 *M*. *alba* var. *atropurpurea* accessions, 124 *M*. *alba* accessions, 46 *M*. *alba* var. *multicaulis* accessions, 4 *M*. *bombycis* accession, 2 *M*. *australis* × *M*. *alba* var. *atropurpurea* accessions, 1 *M*. *australis* accession, 1 *M*. *rubra* accession, 1 *M*. *serrata* accession, and all accessions from *M. alba* var. *indica*, *M. alba* var. *multicaulis* × *M. alba* var. *atropurpurea*, *M. alba* var. *pendula*, *M. alba* × *M. alba* var. *atropurpurea*, *M. mizuho*, *M*. *alba* var. *tortuosa*, and *M. rotundiloba*, meaning no SNPs in these accessions (Appendix A). The second group, characterized by one heterogeneous SNP detected in the ITS1 region, mostly originates from Japan, including 11 *M*. *alba* accessions, 4 *M*. *bombycis* accessions, 3 *M. latifolia* accessions, 2 *M*. *alba* var. *multicaulis* accessions, and 1 *M*. *australis* accession. The third group, with four homogeneous SNPs, comprised seven *M*. *alba* accessions, three *M*. *cathayana* accessions, two *M. macroura* accessions, and one *M. liboensis* accession. The fifth group, containing four heterogeneous SNPs, included two *M. macroura* accessions, two *M*. *cathayana* accessions, one *M. wittiorum* accession, and one *M*. *alba* accession. The sixth group, with 12 heterogeneous SNPs, consisted of three *M. australis* and one *M. australis* × *M*. *alba* var. *atropurpurea* accessions. Four SNPs were detected in the 18S and 26S regions of four *M. mongolica* accessions, while twelve SNPs were identified in two *M*. *wittiorum* accessions. Among fourteen *M. notabilis* accessions, one accession exhibited a unique SNP compared to the other. Two *M*. *yunnanensis* accessions displayed identical sequences to the fourteen *M. notabilis* accessions, except for four heterogeneous SNPs. Six heterogeneous SNPs were detected in five *M. celtidifolia* accessions, while six, seventeen, and three heterogeneous SNPs were identified in three *M. nigra* accessions, three *M*. *rubra* accessions, and one *M*. *serrata* accession, respectively. Given the greater inter- and intraspecific variation in the nrDNA region compared to the ITS region, a phylogenetic tree was reconstructed based on 31 unique sequences converted from nrDNA SNPs of 542 accessions to elucidate the taxonomy of the genus *Morus*. The phylogenetic tree comprised six clades, with robust support for interspecific relationships within *Morus* (Figure 2B). However, the intraspecific relationships exhibited generally lower nodal support. The largest *M. alba* clade (comprising 514 accessions) showed a reticulate phylogenetic structure. Within this clade, *M. alba* accessions were embedded within a polyphyletic assemblage of previously identified species encompassing *M*. *cathayana*, *M*. *australis*, *M. wittiorum*, *M*. *mongolica*, *M. macroura*, *M*. *indica*, *M*. *bombycis*, *M*. *latifolia*, *M*. *mizuho*, *M*. *atropurpurea*, *M. rotundiloba*, *M. multicaulis*, and *M*. *liboensis* (Figure 2B and Appendix A). *Morus cathayana* (five accessions), *M*. *mongolica* (four), and three *M. wittiorum* (three) formed two discrete subclades, while *M. macroura* (four accessions) and *M*. *australis* (five) exhibited polyphyletic distribution across three subclades. Despite being recognized as a distinct species in the *Flora of China*, *Morus liboensis* was a unique node containing *M*. *alba* (seven accessions), *M*. *cathayana* (three), and *M. macroura* (two). The *M. notabilis* clade comprised 14 *M. notabilis* accessions and 2 *M. yunnanensis* accessions. The *M. celtidifolia*, *M. nigra*, *M. rubra*, and *M. serrata* clades included five *M. celtidifolia* accessions, three *M. nigra* accessions, three *M. rubra* accessions, and one *M. serrata* accession, respectively (Figure 2B). Notably, one *M. rubra* accession (ERR4009368) and one *M. serrata* accession (SRR14507007) were clustered within the *M. alba* clade due to the absence of SNPs in their nrDNA (Appendix A).

### 2.3. Characterization of the ITS Region in Morus Based on Cloning and Sequencing

Sixteen publicly available ITS sequences representing separate *Morus* species were selected for comparative analysis from a total of three hundred and seventy-four mulberry ITS sequences (Appendix A) deposited in GenBank. The 5.8S region exhibited lower polymorphism compared to ITS1 and ITS2 (Figure 3), consistent with their SNP profiles (Figure 1A). The *MstI* and *BstEII* restriction sites were identified in the ITS1 region, with an additional *BstEII* site in the ITS2 region, except for in *M. notabilis* (Figure 3). The ITS lengths of *M. notabilis* and *M. alba* were the longest (631 bp) and shortest (611 bp), respectively, among these *Morus* species. A 13 bp InDel was detected in seven *M. alba* accessions out of five hundred and forty-two mulberry accessions. Four out of seven *M. alba* accessions, along with one *M. nigra* accession, were selected for validation of the 13 bp InDel using Sanger clone-based sequencing. Among 32, 21, 17, 15, and 23 clones from *M. nigra*, accession 12, Yunsang-6, Cambodia sang, and Hybrid-n13 (Appendix A), 2, 6, 0, 11, and 9 clones lacked the 13 bp insertion (“CGTGCGCAATGCG”), consistent with the ratios of raw read numbers between the 13 bp InDel and the reference site. Additionally, 26, 18, 15, 11, and 20 diverse ITS sequences were found in 32, 21, 17, 15, and 23 clones from *M. nigra*, accession 12, Yunsang-6, Cambodia sang, and Hybrid-n13, respectively, revealing intragenomic ITS polymorphisms like those in diverse intra-individual ITS sequences (Appendix A), indicating incomplete concerted evolution of the ITS in *Morus*.

### 2.4. ITS-CAPS Analysis of Morus Species

Two restriction enzymes, *BstE*II and *Mst*I, located in the ITS region, enable differentiation among *Morus* species using ITS-CAPS. To facilitate PCR and subsequent analysis, the PCR amplicons were designed to include partial 18S and 26S fragments (41 and 37 bp, respectively). Although the amplicon lengths for six *Morus* species varied by up to 20 bp (Table 4), length polymorphisms were difficult to discern on the 1.5% gel due to minimal nucleotide differences (Figure 4). Clear band differences were observed following the ITS-CAPS assay (Figure 4). Specifically, four bands, including two distinct bands (91 and 449 bp), were detected for *M. alba* and *M. australis*, while two types of bands were observed for *M. nigra* and *M. serrata*. The amplicon of *M. notabilis* remained undigested (Table 4 and Figure 4). Interestingly, *M. australis* accession 12, which harbored a 13 bp heterogeneous InDel, exhibited a unique band pattern resembling both *M*. *alba* and *M. nigra* (Figure 4, and Appendix A). Therefore, *M. alba* and *M. notabilis* could be easily differentiated from the other four *Morus* species using the ITS-CAPS approach, with the 449 bp band specific to *M. alba* and the 709 bp band specific to *M. notabilis*.

## 3. Discussion

### 3.1. Characteristics of nrDNA in Morus: Incomplete Concerted Evolution

#### 3.1.1. The Assembly of nrDNA Regions Remains Challenging

Recent advancements in HiFi and ultra-long sequencing technologies have enabled near-complete assemblies, although gaps remain in challenging regions such as centromeric and pericentromeric repeats and nrDNA arrays [52]. For instance, two gaps were identified in the nrDNA arrays of the recently completed *Nicotiana benthamiana* genome [53]. Even in *Arabidopsis thaliana*, gaps persist in the rDNA region despite improvements [54]. The assembly of the nrDNA region remains challenging due to the presence of hundreds of heterogeneous copies. The gap-free genome of *M. notabilis* revealed approximately 200 heterogeneous nrDNA copies on chromosome 5 [33]. However, the characteristics of rDNA in other mulberry accessions remain largely unexplored, particularly in large sets.

Although numerous ITS sequences of *Morus* have been sequenced for comparative analyses and taxonomic classification [20,42,43,44,55], only a few of partial 18S and 26S sequences have been isolated from *Morus*. This scarcity is likely due to the labor-intensive and costly nature of obtaining full-length 18S or 26S sequences via Sanger sequencing. Intragenomic ITS polymorphism was detected through higher sequencing depth, such as 7–20 clones per accession [43] and 15–32 clones from five mulberry accessions in this study, indicating that ITS variants cannot be effectively investigated using low-depth Sanger sequencing. Consequently, traditional Sanger sequencing is unsuitable for characterizing the nrDNA region at the genome level in *Morus*.

#### 3.1.2. Next-Generation Sequencing Facilitates Characterization of Morus nrDNA

Fortunately, the rapid development of next-generation sequencing technologies has provided large-scale mulberry resequencing data and several chromosome-level mulberry genomes [29,31,32,33], offering opportunities to investigate nrDNA characteristics at the genome level. In this study, the conserved nrDNA sequence from the chromosome-level Chuizhisang genome was used as the reference to determine variants in 542 mulberry accessions. Chuizhisang is commonly used in landscaping due to its weeping canopy [56]. This trait is controlled by a recessive gene [57], suggesting that its nrDNA sequences may be conserved and suitable as a reference. In this study, 536 mulberry accessions exhibited an average depth of over 2625-fold coverage of the nrDNA region, indicating high-copy nrDNA sequences in the *Morus* genome. No InDels were detected in the 18S coding region, confirming that the 18S length in *Morus* was 1808 bp, consistent with the findings in other species such as *Panax* [58], *Brassica* [59], and *Eriobotrya* [60]. Two InDels were identified in the 26S coding region in *Morus*, indicating a 26S length of 3395 bp in *Morus*, except for in *M*. *nigra* and *M. yunnanensis*. Compared to InDels primarily in the ITS region, more than half of the SNPs were detected in the 18S and 26S coding regions, providing valuable polymorphism data for phylogenetic analysis and taxonomic classification of *Morus*. Furthermore, variable numbers and patterns of SNPs and InDels (Figure 1 and Figure 2A) across six *Morus* species revealed high interspecific polymorphism and intraspecific conservation of the nrDNA region. Homogeneous SNPs and InDels indicate the presence of alternative alleles within a sample, while heterogeneous SNPs and InDels suggest the simultaneous presence of reference and alternative alleles. In the *Morus* nrDNA region, many SNPs and InDels were homogeneous, whereas others were heterogeneous (Table 3). Compared to other wild *Morus* species, the SNPs and InDels in *M. alba* samples were predominantly heterogeneous, likely due to its open-pollination nature and long-term artificial selection and hybridization in sericulture. Additionally, despite the 13/16 bp InDel identified in only 35 *Morus* accessions, raw read analysis revealed the presence of both alternative and reference alleles in 539 *Morus* accessions, indicating that this allele was predominantly heterogeneous. Furthermore, 108 ITS sequences cloned from five mulberry accessions also exhibited significant intra-individual polymorphism in *M. alba* and *M. nigra* (Appendix A). The widespread occurrence of heterogeneous SNPs and InDels revealed inter-individual nrDNA polymorphism in *Morus*, indicating non-concerted evolution.

The gap-free genome of *M. notabilis* comprised approximately 200 distinct nrDNA copies [33], consistent with the nrDNA characteristics observed in the 16 *M. notabilis* accessions analyzed in this study. The sixteen *M. notabilis* accessions collected from their native regions displayed heterogeneous SNPs, indicating genetic diversity within the wild populations. In contrast, the chromosome-level assemblies of domesticated *M. alba* [29,30] contained fewer and discontinuous nrDNA sequences, likely due to the absence of sequencing data integrated from Pacbio HIFI and ONT ultra-long reads. In this study, nearly all *M. alba* var. *multicaulis* and *M. alba* var. *atropurpurea* accessions lacked nrDNA polymorphism, representing typical domesticated mulberry accessions developed through prolonged artificial hybridization. Compared to the 16 wild accessions (*M. notabilis*), the 472 domesticated accessions (*M. alba*) exhibited reduced genetic diversity, whereas the remaining *M. alba* accessions with heterogeneous SNPs may represent wild samples, such as *M. mongolica*, *M. australis, M. wittiorum*, and *M. macroura*. Furthermore, significant heterogeneous SNPs were identified in wild species, including *M. celtidifolia*, *M. nigra*, *M. rubra*, and *M. serrata*. Therefore, we propose that heterogeneous SNPs in the nrDNA region serve as a diagnostic marker for wild accessions and can be utilized to differentiate between wild and cultivated mulberry samples.

Approximately half of the SNPs and nearly all of the InDels were detected in the ITS region of *Morus*, confirming its suitability for taxonomic classification [20,42,43,44,55]. A 13 bp InDel in the ITS1 region has been reported in *Morus* [20,42,43] and was also detected in this study. Our results further demonstrated that diverse ITS sequences within a mulberry accession could be detected by sequencing multiple random clones (Appendix A), consistent with previous findings [43]. Therefore, high-throughput sequencing provides a more accurate representation of nrDNA characteristics at the genome level compared to traditional Sanger sequencing.

### 3.2. Taxonomy of the Genus Morus

The genus *Morus* has been classified into six species [25] and five sections [15], with its biogeographic history explored using Hyb-Seq data [26]. *M. alba* var. *atropurpurea* was recently proposed as *M. atropurpurea* [30]. A recent study resolved interspecific relationships within the Asian *Morus* species [28]. These findings highlight the need for further research to refine the taxonomy of *Morus*. Mulberry exhibits a wide range of ploidy levels, from diploid to docosaploid, with 14, 28, 42, 56, 84, 112, and 308 chromosomes [31]. For example, *M. notabilis* is diploid with 14 chromosomes [61], *M. serrata* is polyploid with 56 or 84 chromosomes [16,62], and *M. nigra* is docosaploid with 308 chromosomes [16,63]. The polyploid nature of *Morus* complicates population genetic analyses, as many standard tools developed for diploids are not applicable to polyploids [64]. Consequently, recent population genetic analysis using 134, 167, and 425 mulberry accessions excluded polyploid species such as *M. nigra* and *M. serrata* [29,30,32], resulting in an incomplete taxonomy of the genus. To mitigate the impact of polyploidy on classification, we utilized the chloroplast genome. However, obtaining authentic *M. rubra* samples remains essential for accurate taxonomic classification [25]. The ITS region of nrDNA, a popular molecular marker with high discriminatory power, is commonly employed in plant species identification [65], including *Morus*. Given the numerous SNPs in the 18S and 26S regions, reconstructing the phylogenetic tree using complete nrDNA is crucial for *Morus* taxonomy. Therefore, the 158 high-quality nrDNA SNPs from the 542 mulberry accessions are vital for revising the taxonomy of *Morus*.

The phylogenetic reconstruction using 158 nrDNA SNPs revealed strongly supported interspecific relationships within *Morus*, whereas general low nodal support for intraspecific relationships was found. The obvious differences in bootstrap values in the phylogenetic tree possibly originated from pronounced interspecific divergence coupled with limited intraspecific polymorphisms in the nrDNA region of *Morus* (Figure 2A). For example, a single heterogeneous SNP was detected in fourteen *M. notabilis* accessions, and four heterogeneous SNPs were found between *M. yunnanensis* and *M. notabilis*. Within the *M. alba* clade, 457 accessions lacked polymorphic sites, 21 accessions exhibited a single heterogeneous SNP, and 13 accessions carried four homogeneous SNPs.

Four *Morus rubra* accessions were analyzed in this study. Three of the four *M*. *rubra* accessions exhibited a consistent SNP and InDel pattern, while one accession (ERR4009368) displayed the same SNP and InDel pattern as *M. alba* accessions and was classified within the *M. alba* clade, consistent with recent taxonomy based on complete chloroplast genomes [25]. We proposed that *M. rubra* accession (ERR4009368) may result from misidentification or hybridization between *M. alba* and *M. rubra*. For example, a misidentified *M. rubra* at the Memorial Church of Stanford University was corrected to *M. alba* [66].

*Morus serrata* Roxb, known as Himalayan mulberry, is distributed in the temperate Himalayas [62]. In this study, two *M. serrata* accessions originated from Jilong County, Tibet Autonomous Region, China, and India. Neither SNPs nor InDels were detected in the nrDNA region of the Indian *M. serrata* accession (SRR14507007), whereas the Chinese *M. serrata* accession harbored 44 SNPs and 6 InDels, exhibiting a unique SNP and InDel pattern. Recent phylogenetic analyses have demonstrated that the Indian *M. serrata* accession, sharing complete chloroplast genome identity with 12 *M. alba* var. *indica* accessions [25], was clustered into a clade with *M. alba* var. *indica* [28]. Collectively, these *M. serrata* accessions belongs two separate species, *M. serrata* and *M. alba*.

*Morus celtidifolia* Kunth, native to America, is also known as Texas mulberry or Mexico mulberry. *Morus microphylla* Buckley is a synonym of *M. celtidifolia* [26,67]. In this study, four *M. celtidifolia* accessions and one *Morus microphylla* accession (SRR12282926) were clustered into an *M. celtidifolia* clade, providing molecular evidence that *M. microphylla* is a synonym of *M. celtidifolia*. Black mulberry (*Morus nigra* L.), native to western Asia, is a species with 308 chromosomes [63]. Analysis of the nrDNA region in *M. nigra* identified six and two heterogeneous SNPs and InDels, revealing significant intra-individual polymorphism. This finding was further supported by 26 distinct ITS sequences derived from 32 random clones of a single *M. nigra* accession. The observed polymorphism is likely due to hybrid origin and polyploid nature of *M. nigra* [16].

*Morus notabilis* is a wild mulberry native to Sichuan and Yunnan provinces, China [19]. Among the seventy-two homogeneous SNPs identified across fourteen *M. notabilis* accessions and two *M. yunnanensis* accessions, all were identical except for five heterogeneous SNPs. Similarly, all seven homogeneous InDels were shared between *M. notabilis* and *M. yunnanensis*, with the exception of one heterogeneous InDel unique to *M. yunnanensis*. Consequently, the nrDNA profiles of *M. notabilis* and *M. yunnanensis* were found to be nearly identical. Furthermore, one heterogeneous SNP detected in 13 out of 14 *M. notabilis* accessions confirmed the presence of heterogeneous nrDNA copies, as evidenced by the gap-free genome of *M. notabilis* [33]. The phylogenetic tree based on nrDNA SNPs resolved *M. notabilis* and *M. yunnanensis* accessions as a monophyletic clade, confirming congruence with prior chloroplast genomic and single-copy ortholog analyses [25,32].

Taxonomic revision of the genus *Morus* revealed that *M. alba* is intricate and the largest clade [25,29,30,43,68]. *Morus wittiorum* and *M. macroura* were recognized as separate species due to obvious morphometric differentiation (fruit length: 10–16 cm vs. 6–12 cm) in classical taxonomy [19,22,24]. However, *M. wittiorum* was recently treated as *M. macroura* [28]. In this study, phylogenetic analysis resolved all accessions of *M. wittiorum* and *M. macroura* as a subclade of *M. alba*, consistent with previous molecular taxonomy [20,25,43]. Yunsang-6 was the only *M. macroura* accession harboring the 13 bp homogeneous InDel, whose SNP profile matched *M. alba*, resulting in its classification within the *M. alba* clade. Additionally, accessions Hybrid-n13, 12, and Cambodia sang contained the 13 bp heterogeneous InDel and were classified as *M. alba* due to their SNP similarity with *M. alba* accessions. These four mulberry accessions with the 13 bp InDel were classified as *M. alba*, indicating that variable SNPs in the nrDNA region are key to *Morus* taxonomy. In addition, 27 Japanese mulberry accessions and 36 Husang accessions clustered within the *M. alba* clade, supporting the previous finding that Japanese mulberry germplasm belongs to *M. alba* [25], although some Japanese accessions with long female flower styles were previously classified as *Morus bombysis*. *Morus indica* was initially recognized as a species [18] and later classified as a variety of *M. alba* [24,69]. Here, 19 *M. indica* accessions exhibited the same nrDNA characteristics as the other 438 *M. alba* accessions and were clustered within the *M. alba* clade, further confirming that *M. indica* belongs to *M. alba*. Recently, *M*. *alba* var. *atropurpurea* was proposed as *M. atropurpurea* [30]. However, in this study, nearly 246 *M*. *alba* var. *atropurpurea* accessions exhibited no SNPs (Appendix A), indicating identical nrDNA characteristics between *M*. *alba* var. *atropurpurea* and *M. alba*. Therefore, these 246 accessions were classified as *M*. *alba* var. *atropurpurea*.

### 3.3. A Rapid and Reliable Method to Identify M. alba and M. notabilis

Next-generation sequencing is a powerful tool for detecting genetic variants, including SNPs and InDels. CAPS assays are commonly used to identify differences in restriction fragment lengths caused by SNPs or InDels that create or abolish restriction enzyme recognition sites [46,48,49,50,51]. In this study, we confirmed that *M. alba* and *M. notabilis* could be identified using CAPS analysis of ITS sequences. Mulberry accessions whose ITS amplicons were not digested by *BstE*II and *Mst*I belonged to *M. notabilis*. Accessions with a 449 bp band cut by *BstE*II were classified as *M. alba*, although a 91 bp band cut by *BstE*II was also typical for *M. alba* but often too weak to detect. Most *M. alba* accessions were not cut by the *Mst*I due to low copy numbers of the 13 bp InDel. For *M. australis* accession 12, which had a high ratio of the 13 bp InDel, the 449 bp band was weaker than in other *M. alba* accessions, and the 602 bp band cut by *Mst*I was weaker than in *M. nigra* and *M. serrata* accessions. Accessions without a 449 bp band cut by *BstE*II and with 603 bp or 604 bp bands cut by *Mst*I may belong to *M. nigra*, *M. serrata*, *M. celtidifolia*, or *M. rubra*. Among these, *M. rubra*, native to America, has frequently hybridized with *M. alba*, leading to misidentifications [38,39,40]. In Canada, *M. rubra* is threatened by hybridizations with *M. alba* and is listed as endangered due to its declining population. A recovery strategy has been proposed to protect and restore this species [41,70]. The first step in protecting *M. rubra* is the rapid and accurate identification of *M. rubra*, *M. alba*, and their hybrids. However, distinguishing *M. rubra* from *M. alba* and its hybrids is challenging [71,72,73], necessitating molecular identification methods to screen pure *M. rubra* samples. In theory, using ITS-CAPS, pure *M. rubra* accessions will exhibit a 603 bp band cut by *Mst*I and a 555 bp band cut by *BstE*II, while pure *M. alba* accessions will show a 449 bp band cut by *BstE*II and a 689 bp band cut by *Mst*I. Hybrid accessions may display bands similar to *M. australis* accession 12. Therefore, accessions with 449 bp bands cut by *BstE*II are not pure *M. rubra* accessions, while those without 449 bp bands are candidates for pure *M. rubra*. Recently, CAPS assays were developed for pepper [51], soybean [74], and barley [75] germplasm resources. Since ITS-CAPS can be completed within 4 h, including DNA extraction, PCR, enzymatic digestion, and gel electrophoresis, costly and time-consuming sequencing is unnecessary for identifying pure *M. rubra* samples.

## 4. Materials and Methods

### 4.1. Morus Accessions

A total of 542 mulberry accessions (Appendix A) were selected for this study based on publicly available whole-genome resequencing datasets, representing 16 *Morus* species (Table 1). Raw sequencing data were downloaded from the GenBank and CNGB databases; adapters and low-quality sequences were removed from the raw reads using the FASTP software (version 0.23.5) [76] with default parameters to generate high-quality clean reads for the subsequent analyses. Detailed information on all 542 mulberry accessions is provided in Appendix A.

### 4.2. Variant Calling of nrDNA and the Read Counts for 13/16 Bp Insertions

The 5814 bp nrDNA sequence (Appendix A), encompassing the complete 18S (1808 bp), ITS1 (215 bp), 5.8S (163 bp), ITS2 (233 bp), and 26S (3395 bp) from the Chuizhisang (*M. alba* var. *pendula* Dippel) genome (https://db.cngb.org/search/assembly/CNA0019200/, accessed on 06 October 2022), was selected as the reference sequence for variant calling in the genus *Morus*. Clean reads were aligned to the reference nrDNA sequence using BWA-MEM software (version 2.2.1) [77]. The resulting BAM files were sorted and deduplicated using Picard Tools (version 2.25.0) (http://picard.sourceforge.net, accessed on 10 February 2021). Variants were identified using the GATK HaplotypeCaller pipeline (https://gencore.bio.nyu.edu/variant-calling-pipeline-gatk4/, accessed on 15 August 2024) and filtered with the following parameters: for SNPs, “QD < 2.0, MQ < 40.0, FS > 60.0, SOR > 3.0, QUAL < 30.0, MQrankSum < −12.5, ReadPosRankSum < −8.0”; for InDels, “QD < 2.0, MQ < 40.0, FS > 200.0, SOR > 10.0, QUAL < 30.0, MQrankSum < −12.5, ReadPosRankSum < −8.0”. The R package trackViewer (version 1.44.0) [78] was employed to visualize nrDNA SNPs and InDels, and images were edited using Adobe Photoshop^®^ CC (Adobe Systems Inc., CA, U.S.A.). Samtools (version 1.20) [79] was used to calculate the total reads and generate coverage statistics, including mapped read numbers, coverage, mean depth, mean base quality, and mean mapping quality. Bioawk software (https://github.com/lh3/bioawk, accessed on 25 August 2015) was utilized to quantify reads containing 13/16 bp insertions using the following regular expression: “TCGAAACC. + AATGCGCCCCAACCCC|GGGGTTGGGGCGCATT. + GGTTTCGA” for the insertion and “TCGAAACC. + CCCCAACCCC|GGGGTTGGGG. + GGTTTCGA” for the reference sequence.

### 4.3. nrDNA Variations in 542 Morus Accessions and Phylogenetic Analysis

High-quality SNPs identified in 542 *Morus* accessions were converted into FASTA sequences using the vcf2phylip.py script [80]. Duplicate sequences were removed using seqkit software (version 2.5.0) [81]. To visualize intraspecific polymorphisms in the nrDNA region, differential bases were color-coded to match species names using Adobe Photoshop^®^ CC (Adobe Systems Inc., San Jose, CA, USA). Compared to the ITS region (66 SNPs), the entire nrDNA region (158 SNPs) provided more comprehensive intra- and inter-individual polymorphism data. Therefore, nrDNA SNPs were used to reconstruct a phylogenetic tree of the genus *Morus*. Unique SNP-based FASTA sequences from 542 mulberry accessions were analyzed using the IQ-tree software (version 2.3.5) [82] to generate the maximum likelihood (ML) tree, with the best mode (TNe) selected based on the Bayesian information criterion (BIC). Tree reliability was assessed using the ultrafast bootstrap method with 1000 replicates. The resulting consensus ML tree was visualized and annotated using the Interactive Tree of Life (iTOL) V5 [83] and further edited with Adobe Photoshop^®^ CC (Adobe Systems Inc., San Jose, CA, USA).

### 4.4. ITS Fragment Alignment, Sanger Sequencing, and ITS-CAPS Assay

Sixteen representative ITS sequences from 14 *Morus* species were aligned using the MegAlign software (DNASTAR, Madison, WI, USA) and visualized with the Genedoc program (Pittsburgh Supercomputing Center, PA, USA) to characterize the ITS region. Ten representative mulberry accessions were selected from the Mulberry Germplasm Nursery at Southwest University, China, for Sanger sequencing and ITS-CAPS assay. Genomic DNA was extracted from young leaves using a modified cetyltrimethylammonium bromide protocol [84]. The entire ITS region was amplified using mulberry-specific primers (forward: 5′-GTAACAAGGTTTCCGTAGGTG-3′; reverse: 5′-TAAACTCAGCGGGTAGCC-3′). PCR reactions (20 μL) were performed in a mixture containing 20 ng of genomic DNA, 2 mM PCR buffer, 0.2 mM of each primer, 0.2 mM of each dNTP, and 1 unit of Taq polymerase with 1.25 mM MgCl_2_ (Takara, Dalian, China). The PCR program consisted of an initial denaturation at 98 °C for 2 min, followed by 31 cycles of 98 °C for 10 s, 60 °C for 20 s, and 72 °C for 30 s, with a final extension at 72 °C for 5 min. For Sanger sequencing, PCR-amplified products from five mulberry accessions (*M. nigra*, CNR0342502; *M. australis*, CNR0342505; *M. macroura*, CNR0342481; *M. alba*, CRR443619; and hybrid between *M. australis* and *M. alba*, CNR0342473) were purified, cloned into the pMD19-T vector, and sequenced with over 15 positive clones per accession (Tsingke Biotech Co., Ltd, Chengdu, China). For the ITS-CAPS assay, the ITS PCR fragments amplified from five accessions (*M. alba*, CNR0342488; *M. notabilis*, CNR0342496; *M. serrata*, CNR0342504; *M. nigra*, CNR0342502; and *M. australis*, CNR0342505) were digested with *BstE*II and *Mst*I restriction endonucleases, followed by electrophoretic separation on 1.5% (*w*/*v*) agarose gels.

## 5. Conclusions

In this study, we revealed 158 SNPs and 15 InDels in the nrDNA region of 542 mulberry accessions. And 77, 44, 43, 42, 40, and 21 SNPs and 8, 6, 7, 7, 6, and 3 InDels were identified in *M. notabilis*, *M. serrata*, *M. nigra*, *M. celtidifolia*, *M. rubra*, and *M. alba*, respectively, demonstrating high interspecific polymorphism, intraspecific conservation, and unique characteristics of *Morus* species. The widespread heterogeneous SNPs and InDels in *Morus* indicate intra- and inter-individual nrDNA polymorphism and incomplete concerted evolution. The phylogenetic tree based on nrDNA SNPs divided *Morus* into six clades, corresponding to six species. A rapid and reliable method using CAPS markers in the ITS region was developed to distinguish *M. alba* and *M. notabilis* within *Morus* without clone-based sequencing or comparative phenotypic analysis. In summary, the present study provides detailed insights into the nrDNA characteristics of *Morus* and classifies the genus into six species, advancing our understanding of mulberry taxonomy.

## Figures and Tables

**Figure 1 plants-14-02570-f001:**
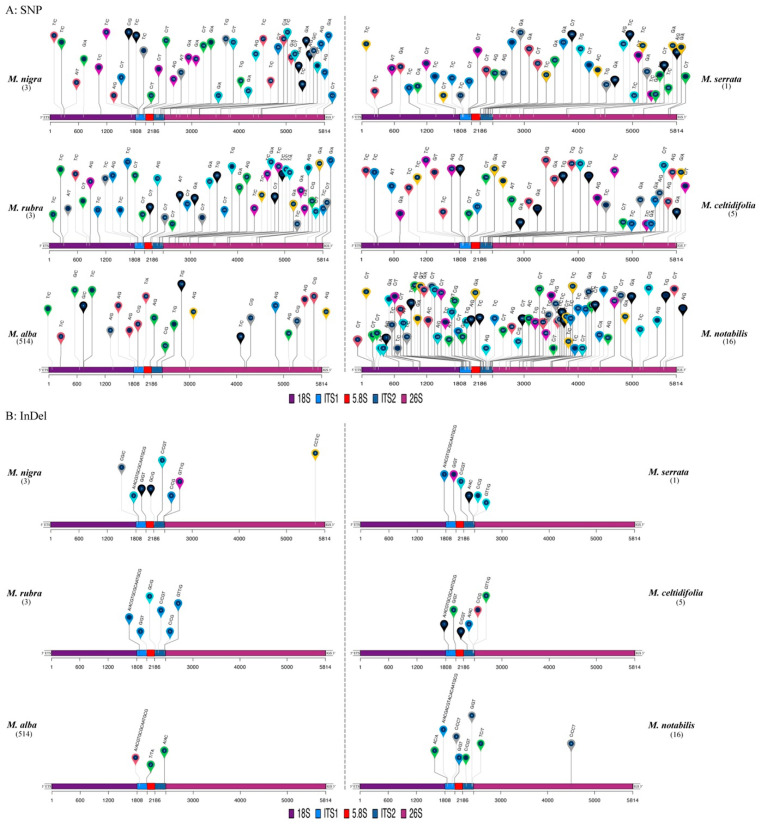
Characteristics and patterns of SNPs and InDels in the nrDNA regions of 542 mulberry accessions corresponding to six *Morus* species. (**A**) SNP distribution within the nrDNA region of the *Morus* genus; (**B**) InDel distribution within the nrDNA region of the *Morus* genus. Colored circles and numbers within circles indicate the loci and counts of SNPs and InDels in the nrDNA regions of the *Morus* genus, respectively. SNPs and InDels are denoted by reference bases and alternative bases separated by a slash (e.g., T/C: T represents the reference bases, and C represents alternative bases; CG/C: CG represents the reference bases, and C represents the alternative bases). Numbers in parentheses indicate the counts of mulberry accessions.

**Figure 2 plants-14-02570-f002:**
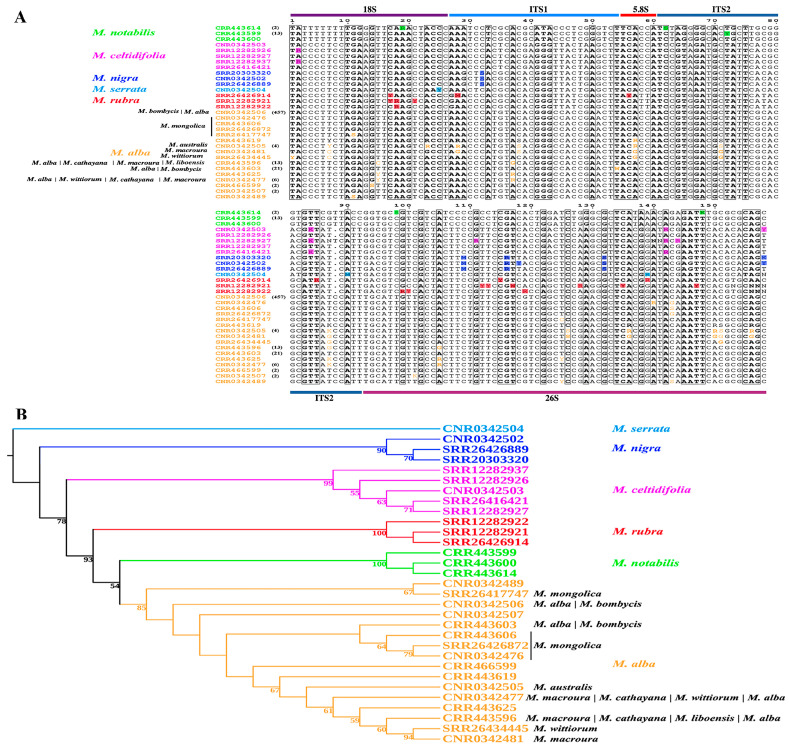
Comparison of the 158 SNPs identified from the nrDNA region of 542 Morus accessions (Appendix A) and their phylogenetic relationships. (**A**) Inter- and intraspecific polymorphism of the nrDNA region in *Morus*. Intraspecific variations are color-coded to match species names. Numbers in parentheses indicate the count of repetitive sequences with identical SNP data. Colored boxes represent the distribution of SNPs across the 18S, ITS, 5.8S, ITS2, and 26S regions. According to the nucleotide ambiguity code (IUPAC), R = A or G, Y = C or T, K = G or T, M = A or C, S = G or C, and W = A or T. Specifically, the R symbol indicates a heterozygous biallelic site with co-occurrence of adenine (A) and guanine (G) alleles. (**B**) Maximum likelihood phylogenetic tree constructed using the IQ-tree software (version 2.3.5) with the best mode TNe. Bootstrap values exceeding 50% are displayed above the nodes. The vertical bar (|) denotes logical conjunction (“and”).

**Figure 3 plants-14-02570-f003:**
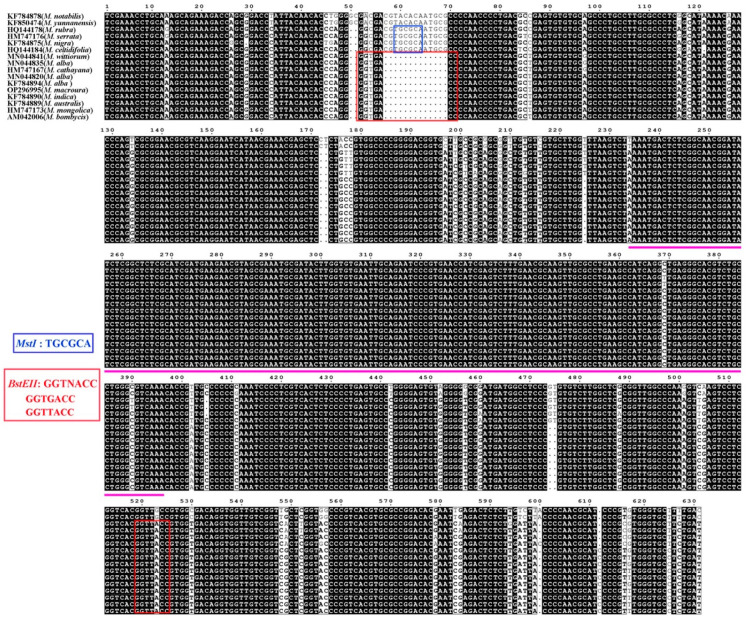
Comparison of ITS sequences and restriction enzyme sites in *Morus* species. Identical sequences are denoted by a black background, while variable sequences are highlighted with a white background. Gaps introduced to optimize alignment are marked with black dots. The region between the pink lines represents the 5.8S sequence. Blue and red rectangles indicate the recognition sites for *Mst*I (TGCGCA) and *BstE*II (GGTNACC), respectively.

**Figure 4 plants-14-02570-f004:**
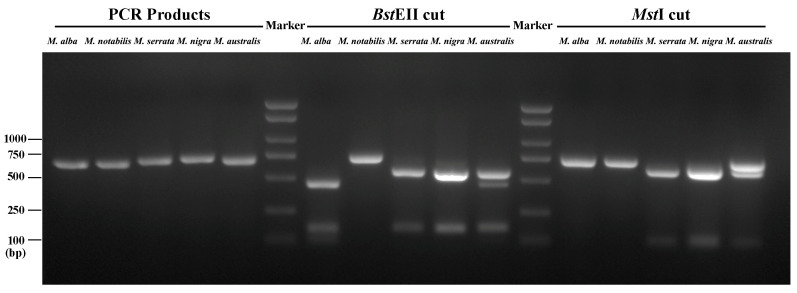
Amplicon and ITS–CAPS banding patterns for four *Morus* species. The following accessions were analyzed: *M. alba* (Chuizhisang, CNR0342488), *M. notabilis* (Chuansang, CNR0342496), *M. serrata* (JiLongSang, CNR0342504), *M. nigra* (black mulberry, CNR0342502), and *M. australis* (12, CNR0342505). A DNA marker was included for reference.

**Table 1 plants-14-02570-t001:** Sixteen *Morus* species and sample sizes by species.

Species	Sample Size	Species	Sample Size
*M. alba*	476	*M. macroura*	4
*M. notabilis*	16	*M. wittiorum*	3
*M. bombycis*	8	*M. latifolia*	3
*M. australis*	5	*M. nigra*	3
*M. celtidifolia*	5	*M. mizuho*	2
*M. cathayana*	5	*M. serrata*	2
*M. mongolica*	4	*M. rotundiloba*	1
*M. rubra*	4	*M. liboensis*	1

Note: *M. alba* comprised 476 accessions representing eight taxonomic groups: *M. alba* var. *atropurpurea* (*n* = 246), *M. alba* (*n* = 146), *M. alba* var. *multicaulis* (*n* = 49), *M. alba* var. *indica* (*n* = 19), *M. alba* var. *multicaulis* × *M. alba* var. *atropurpurea* (*n* = 6), *M. alba* var. *pendula* (*n* = 4), *M. australis* × *M. alba* var. *atropurpurea* (*n* = 3), *M. alba* × *M. alba* var. *atropurpurea* (*n* = 2), and *M. alba* var. *tortuosa* (*n* = 1).

**Table 2 plants-14-02570-t002:** The number of SNPs and InDels in the nrDNA region of *Morus*.

Species	Accession Number	SNP Number in 18S	SNP Number in ITS	SNP Number in 26S	InDel Number in 18S	InDel Number in ITS	InDel Number in 28S
*M. nigra*	3	6	17	20	0	7	1
*M. serrata*	1	8	19	17	0	6	0
*M. rubra*	3	8	16	23	0	6	0
*M. celtidifolia*	5	6	19	17	0	6	0
*M. notabilis*	16	11	34	32	0	7	1
*M. alba*	57	6	7	10	0	3	0

**Table 3 plants-14-02570-t003:** Distribution of homogeneous and heterogeneous SNPs and InDels in *Morus*.

Species	Homogeneous SNP Number	Heterogeneous SNP Number	Homogeneous InDel Number	Heterogeneous InDel Number
*M. nigra*	37	6	6	2
*M. serrata*	41	3	6	0
*M. rubra*	30	17	6	0
*M. celtidifolia*	36	6	6	0
*M. notabilis*	72	5	7	1
*M. alba*	0	23	0	3

**Table 4 plants-14-02570-t004:** Lengths (bp) of PCR products and DNA fragments digested by *BstE*II and *Mst*I.

Species	Length of PCR Products	Length of DNA Fragments Digested by Enzymes	
		*BstE*II	*Mst*I
*M. alba*	689|702	(91/149/449)|(149/553)	689|(100/602)
*M. celtidifolia*	704	149/555	100/604
*M. rubra*	703	148/555	100/603
*M. notabilis*	709	709	709
*M. serrata*	703	148/555	100/603
*M. nigra*	702	148/554	99/603

Symbol conventions: the vertical bar (|) denotes logical disjunction (“or”), whereas the forward slash (/) indicates logical conjunction (“and”).

## Data Availability

The genomic data for 542 mulberry accessions were obtained from the CNGB Sequence Archive of the China National Database (CNGBdb) and the National Center for Biotechnology Information. Specifically, mulberry accessions CNR0342473-CNR0342508, CRR443596-CRR443626, and CRR466515-CRR466804 were retrieved from CNGBdb with the number CNP0001407 and the project accessions PRJCA008608 and PRJCA009071, respectively. Accessions SRR10770733-SRR10770866 and SRR14506989-SRR14507009 were sourced from the NCBI under project accessions PRJNA597170 and PRJNA728807, respectively. Additionally, 374 mulberry ITS sequences were downloaded from GenBank.

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
