# Peer review of "Extensive nrDNA Polymorphism in Morus L. and Its Application"

_plants, 2025, doi:10.3390/plants14162570_

Round 1
Reviewer 1 Report
Comments and Suggestions for Authors
The subject matter of the manuscript is in line with the scope of Plants, but the quality of the manuscript, despite its strengths, is insufficient to warrant publication. Without going into the strengths, I would like to draw the authors' attention to the most important (and fundamental) weaknesses.
- The title of the manuscript does not correspond to the content. The results of the study have no application to taxonomy in the classical sense. The manuscript contains some general reflections on taxonomy, but they are very far from real taxonomic assessments.
- The introduction is seriously lacking a taxonomic overview. A taxonomic review of the genus Morus cannot be based on general considerations, but must include a detailed analysis. I just doubt whether it is needed here, as the article (see note 1) does not deal with the actual taxonomy.
- Overall, the introduction is very short and lacks information about the genus Morus.
- There is a lack of a clearly formulated hypothesis and research questions. An annotation at the end of the introduction explains what has been done, but it is not clear what the purpose of the research was and what it was intended to determine. The whole introduction needs to be restructured.
- The authors claim to have refined the taxonomy of Morus, but the methodology is silent, and the results do not address how genetic results correlate with morphological characters. In general, it is very questionable to base a study on sequences extracted from databases and to create a taxonomy without even knowing how real organisms (in this case, Morus species) differ (and whether they do) in terms of morphological characters. I agree that research is possible, but then the results must be interpreted only at the genetic level. Taxonomic studies without reference to type material and morphological characters are akin to fortune telling.
- I have serious reservations about the illustrations. I understand that they are not easy to draw and take up a lot of space, but look at it from the reader's point of view! How could anyone understand and interpret part A of Figure 1? First of all, the quality is unsatisfactory because of the smallness of the entries. Would it not be better to make the illustration larger so that it is clearly legible? Or is it only the differences that need to be highlighted?
- Conclusions are not conclusions, but summaries. Conclusions should contain the main message of the study. The etymology and semantic meaning of the word "Conclusions" itself requires the presentation of a key concluding statement or statements.
Author Response
Response to Reviewer:
We appreciate the time and effort taken by the reviewer to evaluate our manuscript. However, we respectfully disagree with several critiques that appear to misinterpret the scope, methodology, and contributions of our study. And we feel compelled to clarify some misunderstandings that suggest a potential misalignment between your expectations and the actual scope of our study.
Here is a point-by-point response to your comments and concerns, clarifying key misunderstandings. We hope that our explanation will meet with approval.
Comments 1: The subject matter of the manuscript is in line with the scope of Plants, but the quality of the manuscript, despite its strengths, is insufficient to warrant publication. Without going into the strengths, I would like to draw the authors' attention to the most important (and fundamental) weaknesses.
The title of the manuscript does not correspond to the content. The results of the study have no application to taxonomy in the classical sense. The manuscript contains some general reflections on taxonomy, but they are very far from real taxonomic assessments.
The introduction is seriously lacking a taxonomic overview. A taxonomic review of the genus Morus cannot be based on general considerations, but must include a detailed analysis. I just doubt whether it is needed here, as the article (see note 1) does not deal with the actual taxonomy.
Overall, the introduction is very short and lacks information about the genus Morus.
There is a lack of a clearly formulated hypothesis and research questions. An annotation at the end of the introduction explains what has been done, but it is not clear what the purpose of the research was and what it was intended to determine. The whole introduction needs to be restructured.
Response 1: The reviewer suggests that the title does not match the content and that the results have no application to classical taxonomy. We respectfully disagree. Our manuscript does not claim to present a "classical taxonomic assessment" or revise Morus taxonomy in the traditional morphological sense, as you seem to assume. The title reflects our work on genome-level nrDNA characteristics of Morus species, with implications for phylogenetic relationships—a distinction we believe was clear in the text, which was consistent with your comments “The manuscripts contain some general reflections on taxonomy”. The genus Morus (mulberry) exhibits significant taxonomic inconsistency, with reported species numbers ranging from 5 to 35 due to discrepancies in morphological classification. However, population genetic analyses have demonstrated that traditional morphology-based taxonomy does not accurately reflect the true phylogenetic relationships within the genus. Recent studies further support the existence of distinct Morus species, underscoring the need for a more reliable taxonomic framework. Nuclear ribosomal DNA (nrDNA) consists of hundreds of tandemly repeated units, each containing three conserved coding regions (18S, 5.8S, and 26S rRNA genes) separated by internal transcribed spacers (ITS1 and ITS2) and flanked by external transcribed spacers (ETS) and intergenic spacer regions (IGS). While the ITS region has been widely utilized in phylogenetic studies for species discrimination, the complete nrDNA region harbors substantially greater sequence variability, offering enhanced resolution for phylogenetic inference. Despite the availability of numerous Morus ITS sequences, the structural and evolutionary characteristics of the entire nrDNA region—particularly at the genomic level and across diverse mulberry accessions—remain poorly characterized. To address this gap, the present study aims to (1) comprehensively characterize the nrDNA region in the Morus genus and (2) evaluate its utility in refining the taxonomy of Morus species. In the section 2 of Introduction (Lines 48-65 in the manuscript), we reviewed the taxonomy of genus Morus with almost all the papers published from 1753 to 2024. According to your suggestion, we restructured the Introducion section in the revised manuscript.
Comments 2: The authors claim to have refined the taxonomy of Morus, but the methodology is silent, and the results do not address how genetic results correlate with morphological characters. In general, it is very questionable to base a study on sequences extracted from databases and to create a taxonomy without even knowing how real organisms (in this case, Morus species) differ (and whether they do) in terms of morphological characters. I agree that research is possible, but then the results must be interpreted only at the genetic level. Taxonomic studies without reference to type material and morphological characters are akin to fortune telling.
Response 2: In section 2.3 of Materials and Methods, the methodology of taxonomy of Morus was described. We respectfully point out that this methodology follows standard practices in molecular systematics. The reviewer questions the validity of using sequences from public databases and criticizes the lack of morphological correlation. We respectfully point out that these publicly available data have been employed to comparatively analyze for taxonomic studies of Morus, in general, these mulberry data were classified by their authors based on morphological characters. In present study, we accept their morphological taxonomy (Table 1 and Table S1). According to our findings of nrDNA characteristics, high interspecific polymorphism, intraspecific conservation and unique characteristics of Morus species, we proposed that M. rubra accession (ERR4009368) and Indian M. serrata accession (SRR14507007) were misidentified. Futhermore, M. alba var atropurpurea recently proposed as M. atropurpurea species, M. wittiorum, M. macroura, M. australis, M. cathayana, M. mongolica and M. liboensis accessions displayed identical SNP and InDel pattern as M. alba accessions and also were clustered into M. alba clade, consistent with recent taxonomy based on complete chloroplast genomes. As a result, this study also supported that traditional morphology-based taxonomy does not accurately reflect the true phylogenetic relationships within the genus Morus.
Comments 3: I have serious reservations about the illustrations. I understand that they are not easy to draw and take up a lot of space, but look at it from the reader's point of view! How could anyone understand and interpret part A of Figure 1? First of all, the quality is unsatisfactory because of the smallness of the entries. Would it not be better to make the illustration larger so that it is clearly legible? Or is it only the differences that need to be highlighted?
Response 3: We appreciate the reviewer’s feedback regarding the clarity of Figure 1. In order to visualize nrDNA characteristics of SNPs and InDels, the R package trackViewer was employed then edited by Photoshop to draw the Figure 1. According to the Figure 1, it is clear that SNPs and InDels display unique patterns and correspond to six Morusspecies including M. nigra, M. serrata, M. rubra, M. celtidifolia, M. alba, and M. notabilis. These differences (unique patterns in each species) were displayed and highlighted, revealing interspecific diversity and intraspecific conservation. Furthermore, we also have submitted high-resolution versions of Figure 1.
Comments 4: Conclusions are not conclusions, but summaries. Conclusions should contain the main message of the study. The etymology and semantic meaning of the word "Conclusions" itself requires the presentation of a key concluding statement or statements.
Response 4: We agree that the original "Conclusions" section was more of a summary than a set of key findings. In the revised version, we have rewritten this section to highlight the main contributions and implications of our study, ensuring it reflects the core message of the paper.
Once again, we thank the reviewer for their insightful and constructive comments, which have significantly improved the quality of our manuscript. All changes made in the revised manuscript are clearly marked and highlighted for ease of review.
Reviewer 2 Report
Comments and Suggestions for Authors
This study takes existing nuclear ribosomal DNA sequence for a large sample of several Morus species and analyses them in depth to better understand intra-individual variation, and to develop more convenient markers for species identification. This study provides incremental insights into rDNA evolution in this genus as well as more general insights into the limits of concerted evolution at rDNA loci.
The introduction starts by explaining the study system and current state of knowledge. The challenges of Morus taxonomy and limits to current understanding are summarised well. The strong introduction structure leads to clear study aims and approaches statements.
The methods generally gives a very good level of concise detail allowing repeatability.
The opening results section describing SNP variation across samples is very long. Consider breaking up into more paragraphs, for example by SNP/InDel, for better readability. The final results section about cloning results also mixes public data analysis, which is confusing to follow. Some rewriting here would help.
The discussion develops the rDNA results with more consideration of challenges in the field and how the approaches of this study can contribute to addressing them. The discussion organises topics well but more subheadings would help the reader follow the different topics better. The discussion shows depth of integration with related literature. The breakdown of species/clade designations is a useful summary of the phylogenetic study. Potential uses of the new CAPS marker for M rubra conservation add practical interest. Perhaps some relevant examples from other species cold be included in each part of the discussion to show the broader applicability of the approaches.
Specific comments
L93-94 Specify that this method involved the development of a CAPS marker.
L100-101 Reference or specify the read cleaning thresholds used.
L103 Reference the data availability statement for the sources of these public datasets.
L150-151 Reference or specify the modified CTAB protocol used.
L161-162 Briefly describe how the RE sites variants were identified and chosen. What tools were used?
L221-223 Does this result mean that the 13/16 bp Indel is nearly ubiquitous. Why might it not have been scored in so many samples? Is this a read mapping issue?
L223-225 I suggest to dial back this interpretation to state incomplete concerted evolution rather than non-concerted evolution as most variation is homogenous within samples.
L299 The header for this section concerns cloning and sequencing but the text refers to GenBank sequences. Please clarify this confusion.
L329 Typo change "rations" to "ratios"
L508 Is there a word missing after "intricate"?
Author Response
Response to Reviewer:
We sincerely appreciate the positive assessment of the study’s significance, the structure of the introduction, and the clarity of the methods section. We fully agree with your suggestions for improving the manuscript. A revised manuscript with the corrected sections highlighted in red is provided for review and editing purposes.
Here is a point-by-point response to your comments and concerns. We hope that our explanation will meet with approval.
General Comments
Comments 1: This study takes existing nuclear ribosomal DNA sequence for a large sample of several Morus species and analyses them in depth to better understand intra-individual variation, and to develop more convenient markers for species identification. This study provides incremental insights into rDNA evolution in this genus as well as more general insights into the limits of concerted evolution at rDNA loci.
Response 1: Thank you for this accurate and generous summary of our study. We are pleased that the reviewer recognizes the value of our work in contributing to the understanding of rDNA evolution and species identification in Morus.
Comments 2: The introduction starts by explaining the study system and current state of knowledge. The challenges of Morus taxonomy and limits to current understanding are summarised well. The strong introduction structure leads to clear study aims and approaches statements.
Response 2: We are grateful for the reviewer’s positive assessment of the introduction. We have maintained the structure in the revised manuscript to ensure clarity of purpose and context.
Comments 3: The methods generally gives a very good level of concise detail allowing repeatability.
Response 3: Thank you for your recognition of the methodological clarity. We have ensured that all procedures remain clearly described to support reproducibility.
Comments 4: The opening results section describing SNP variation across samples is very long. Consider breaking up into more paragraphs, for example by SNP/InDel, for better readability. The final results section about cloning results also mixes public data analysis, which is confusing to follow. Some rewriting here would help.
Response 4: We appreciate the suggestion and have restructured result 3.1 section into three paragraphs to improve readability and flow. Additionally, we have revised this section to clearly separate the cloning results by deletion of public data analysis (Lines333-353), improving the clarity of the presentation.
Comments 5: The discussion develops the rDNA results with more consideration of challenges in the field and how the approaches of this study can contribute to addressing them. The discussion organises topics well but more subheadings would help the reader follow the different topics better. The discussion shows depth of integration with related literature. The breakdown of species/clade designations is a useful summary of the phylogenetic study. Potential uses of the new CAPS marker for M rubra conservation add practical interest. Perhaps some relevant examples from other species cold be included in each part of the discussion to show the broader applicability of the approaches.
Response 5: Thank you for your detailed feedback. We have added subheadings in the discussion to better organize the content and guide the reader (Lines 387, 408). Furthermore, we have incorporated relevant examples from other plant species (Lines 604-605) to illustrate the broader applicability of CAPS, as suggested.
Specific comments
Comments 6: L93-94 Specify that this method involved the development of a CAPS marker.
Response 6: Thank you for pointing this out. We have revised the text to explicitly state that the method involved the development of a CAPS marker (Line 100).
Comments 7: L100-101 Reference or specify the read cleaning thresholds used.
Response 7: Thank you for your careful review. The Fastp software was run with default parameters to generate high-quality clean reads (Lines 108-109).
Comments 8: L103 Reference the data availability statement for the sources of these public datasets.
Response 8: Thank you for pointing this out. We have added a reference to the Data Availability Statement to clarify the sources of the public datasets used (Lines 645-653).
Comments 9: L150-151 Reference or specify the modified CTAB protocol used.
Response 9: Thank you for your careful review. We have added a reference to the specific modified CTAB protocol used for DNA extraction.
Comments 10: L161-162 Briefly describe how the RE sites variants were identified and chosen. What tools were used?
Response 10: Thank you for pointing this out. PCR were used to amplify the ITS fragments.
Comments 11: L221-223 Does this result mean that the 13/16 bp Indel is nearly ubiquitous. Why might it not have been scored in so many samples? Is this a read mapping issue?
Response 11: Thank you for your careful review. In this study, although 539 samples containing 13/16-bp raw reads, the 13/16-bp InDel were called in only 35 samples because most of samples contained a few 13/16-bp raw reads failed to be called as InDel. In the revised manuscript, the second “InDel” were corrected to “insertion” for clarity.
Comments 12: L223-225 I suggest to dial back this interpretation to state incomplete concerted evolution rather than non-concerted evolution as most variation is homogenous within samples.
Response 12: We appreciate this suggestion and have revised the interpretation accordingly, using the term "incomplete concerted evolution" to more accurately reflect our findings in the revised manuscript.
Comments 13: L299 The header for this section concerns cloning and sequencing but the text refers to GenBank sequences. Please clarify this confusion.
Response 13: Thank you for catching this inconsistency. We have revised the header and the text to clearly reflect that this section by deletion of public data analysis (Lines333-353), improving the clarity of the presentation.
Comments 14: L329 Typo change "rations" to "ratios"
Response 14: Thank you for pointing this out. The "rations" has been corrected to “ratios.
Comments 15: L508 Is there a word missing after "intricate"?
Response 15: Thank you for pointing this out. We have reviewed the sentence and revised it for clarity.
Once again, we thank the reviewer for their insightful and constructive comments, which have significantly improved the quality of our manuscript. All changes made in the revised manuscript are clearly marked and highlighted for ease of review.
Round 2
Reviewer 1 Report
Comments and Suggestions for Authors
The manuscript has been corrected after the first round of reviews, but it is unfortunate that many of the comments were not taken into account and the answers are not based on scientific arguments, but repeat what has already been written in the text and criticized. I will therefore try to explain further what has led to this criticism.
- The content of the manuscript did not, and still does not, correspond to the title. There is no taxonomy in the article. All the more so as this study cannot be called a taxonomic revision, as it is stated at the end of the introduction (lines 85-86). A taxonomic revision involves critically and comprehensively reassessing the taxonomy of a group of organisms (e.g. a genus, family or species complex) by integrating morphological, genetic, ecological and geographical data in order to clarify classification, synonymy and relationships. This manuscript, at best, touches on one aspect of taxonomy, the search for phylogenetic relationships among species of the genus Morus.
- This manuscript, at best, touches on one aspect of taxonomy, the search for phylogenetic relationships among species of the genus Morus. I have written that the introduction does not cover all the problems addressed, but it is not complete and the essential deficiencies have not been corrected. How to understand the statement from the taxonomy in nomenclature perspective (lines 50-52) "Morus mongolica, M. cathayana, M. australis, M. wittiorum, M. indica, M. atropurpurea, M. bombycis, and M. macroura were proposed as M. alba species using the ITS marker and chloroplast genomes"? Your statements contradict all the principles of taxonomy. I have looked at the sources you cite, but they do not deal with taxonomy and they do not rely on any representative material, so they cannot be considered taxonomic (the authors of the cited papers do not consider them taxonomic). I therefore recommend once again to rethink the title of the paper so that it reflects the content and not what is not there.
- Subject matter errors must be corrected. I cannot list them all, as the introduction and discussion would have to be rewritten. It is not the reviewer's responsibility to provide the authors with the basics of taxonomy, nomenclature and systematics. What taxonomic group is "higher plants"?
- How can the authors make sure that the information used in this study is actually about the specified Morus species and not misidentified specimens? Can the authors be sure that the specimens examined do not include hybrids and artificially derived varieties (cultivars) that are abundant in the genus? What do the results actually show? Is it the taxonomy, the phylogeny or the simple genetic diversity of plants of the genus Morus?
- I am by no means denying that the research is important and deserves to be published, but it is necessary to give things their real names, to interpret the information objectively, and not to draw conclusions that are not possible to draw from the available material and from the research methods used.
Author Response
Response to Reviewer:
Dear Reviewer,
Thank you very much for your detailed and constructive comments on our manuscript. We greatly appreciate the time and effort you have devoted to providing us with such insightful feedback, which has been invaluable in helping us improve the quality of our work. We have carefully considered all your suggestions and have made extensive revisions to address the concerns raised.
Revision of the Title
As strongly recommended by you, we have revised the title of the manuscript by replacing "taxonomic" with "its" to accurately reflect the core content of our study. This revision is intended to ensure that the title aligns precisely with our research focus on molecular data (nrDNA) rather than formal taxonomic revision, and we believe it now better conveys the nature of our work.
Clarification on the Scope of Our Study
We fully acknowledge your point that our study does not constitute a taxonomic revision of the genus Morus. As previous studies have already indicated that traditional morphological classification alone cannot accurately reflect the taxonomy of Morus, our research is based on molecular data, specifically phylogenetic analyses and investigations into the characteristics of nrDNA in Morus. Our primary goal is not to revise the taxonomic system of Morus but to provide new insights into its classification through the analysis of our phylogenetic trees and the features of nrDNA.
We have carefully revised the introduction and discussion sections to clarify this focus, ensuring that our claims are consistent with the scope of our research. We have removed or rephrased statements that might have implied a broader taxonomic revision and have emphasized that our work contributes to the understanding of phylogenetic relationships and genetic characteristics, which can inform taxonomic studies.
Regarding how we confirmed which species the samples we used belonged to, we classified them according to the morphological characters of the taxonomic list of mulberry plants in Flora of China (2003), and the provenance of all the samples we included in the Appendix material detailing which study these materials came from, which ensured their reliability.
We have revised the conclusions to ensure they are strictly based on our data and methods, avoiding any overstatements or interpretations that go beyond the scope of our findings.
Once again, we are deeply grateful for your critical and helpful comments, which have significantly improved our manuscript. We believe these revisions have addressed the key issues you raised, and we hope the revised version now meets the standards for publication.
Sincerely,
Qiwei
Round 3
Reviewer 1 Report
Comments and Suggestions for Authors
After the second round of reviews, the manuscript has been significantly improved and the most significant shortcomings have been eliminated. Nevertheless, there are still some technical or editorial shortcomings.
- There are many missing periods after "var." (they are necessary). For example, line 55.
- Some scientific names are not written in italics (line 55).
- Not all plant names are accurate (lines 79-80). For example, what is "strawberry"? What species? Is it Brassica napus or its seeds (rapeseed)? What is pepper? Is it Piper or Capsicum? Be precise! Write scientific and accurate names of organisms.
- What is the meaning of the unfinished sentence (line 348)? Is it a third-level heading? Was it forgotten to be completed?
- If the unfinished sentence in line 367 is a third-level heading, why does it not have a number and heading format?
- In line 461, the Internet source is not cited in accordance with the journal's requirements. See the requirements for citing literature.
Author Response
Dear Reviewer,
Thank you very much for your careful review and valuable suggestions on our manuscript. We sincerely appreciate the reviewers’ meticulous feedback and the opportunity to further refine our manuscript. We have carefully addressed all the technical and editorial issues you pointed out. A revised manuscript with the corrected sections highlighted in red is provided for review and editing purposes.
Here is a point-by-point response to your comments and concerns. We hope that our explanation will meet with approval.
Comments 1: There are many missing periods after "var." (they are necessary). For example, line 55.
Response 1: We apologize for this oversight. All instances of "var" have been corrected to "var." throughout the manuscript (Lines 55, 248, 426).
Comments 2: Some scientific names are not written in italics (line 55).
Response 2: Thank you for your detailed feedback. We have performed a full-text audit to ensure compliance with taxonomic formatting standards. For example, "atropurpurea" in lines 55 and 426 have been corrected to "atropurpurea".
Comments 3: Not all plant names are accurate (lines 79-80). For example, what is "strawberry"? What species? Is it Brassica napus or its seeds (rapeseed)? What is pepper? Is it Piper or Capsicum? Be precise! Write scientific and accurate names of organisms.
Response 3: Thank you for pointing this out. We have revised the imprecise plant names mentioned in lines 79-80: "Strawberry" has been specified as "Fragaria × ananassa", "Flax" has been corrected to "Linum usitatissimum", "Rapeseed" is clearly indicated as "Brassica napus", and "Pepper" has been corrected to "Capsicum annuum".
Comments 4: What is the meaning of the unfinished sentence (line 348)? Is it a third-level heading? Was it forgotten to be completed? If the unfinished sentence in line 367 is a third-level heading, why does it not have a number and heading format?
Response 4: We regret these editorial oversights. Both headings have been reformatted to comply with the journal’s style guide. Line 348: Revised to "4.1.1 The assembly of nrDNA regions remains challenging". Line 367: Revised to "4.1.2 Next-Generation Sequencing facilitates characterization of Morus nrDNA ".
Comments 5: In line 461, the Internet source is not cited in accordance with the journal's requirements. See the requirements for citing literature.
Response 5: Thank you for your careful review. We have revised the citation of the Internet source to strictly adhere to the journal's requirements for citing online materials.
Once again, we thank the reviewer for their insightful and constructive comments, which have significantly improved the quality of our manuscript. All changes made in the revised manuscript are clearly marked and highlighted for ease of review.